Gaining more from doing less? The effects of a one-week deload period during supervised resistance training on muscular adaptations

Coleman Max 1
Burke Ryan 1
Augustin Francesca 1
Piñero Alec 1
Maldonado Jaime 1
Fisher James P. 2
Israetel Michael 1
Androulakis Korakakis Patroklos 1
Swinton Paul 3
Oberlin Douglas 1
Schoenfeld Brad J. bradschoenfeldphd@gmail.com 1
1 Applied Muscle Development Laboratory, City University of New York, Herbert H. Lehman College , Bronx , United States of America
2 Solent University , Southampton , United Kingdom
3 Robert Gordon Univesity , Aberdeen , United Kingdom
Keogh Justin
Electronic publication date: 2024 Jan 22
Publication date: 2024
Volume: 12
Electronic Location ID: e16777
Received 2023 Jun 29; Accepted 2023 Dec 18
Copyright: ©2024 Coleman et al.
Copyright year: 2024
Copyright holder: Coleman et al.
License: This is an open access article distributed under the terms of the Creative Commons Attribution License, which permits unrestricted use, distribution, reproduction and adaptation in any medium and for any purpose provided that it is properly attributed. For attribution, the original author(s), title, publication source (PeerJ) and either DOI or URL of the article must be cited.
License URL: https://creativecommons.org/licenses/by/4.0/

Keywords: Detraining, Hypertrophy, Strength, Muscle endurance, Resensitize

Funding: Renaissance Periodization, LLC This study was supported by a grant from Renaissance Periodization, LLC. Renaissance Periodization had a role in the study design and preparation of the manuscript.

==============================
Background

Based on emerging evidence that brief periods of cessation from resistance training (RT) may re-sensitize muscle to anabolic stimuli, we aimed to investigate the effects of a 1-week deload interval at the midpoint of a 9-week RT program on muscular adaptations in resistance-trained individuals.

Methods

Thirty-nine young men (n = 29) and women (n = 10) were randomly assigned to 1 of 2 experimental, parallel groups: An experimental group that abstained from RT for 1 week at the midpoint of a 9-week, high-volume RT program (DELOAD) or a traditional training group that performed the same RT program continuously over the study period (TRAD). The lower body routines were directly supervised by the research staff while upper body training was carried out in an unsupervised fashion. Muscle growth outcomes included assessments of muscle thickness along proximal, mid and distal regions of the middle and lateral quadriceps femoris as well as the mid-region of the triceps surae. Adaptions in lower body isometric and dynamic strength, local muscular endurance of the quadriceps, and lower body muscle power were also assessed.

Results

Results indicated no appreciable differences in increases of lower body muscle size, local endurance, and power between groups. Alternatively, TRAD showed greater improvements in both isometric and dynamic lower body strength compared to DELOAD. Additionally, TRAD showed some slight psychological benefits as assessed by the readiness to train questionnaire over DELOAD.

Conclusion

In conclusion, our findings suggest that a 1-week deload period at the midpoint of a 9-week RT program appears to negatively influence measures of lower body muscle strength but has no effect on lower body hypertrophy, power or local muscular endurance.

Introduction

A compelling body of evidence indicates that resistance training (RT) can promote appreciable increases in muscle size and strength (Kraemer, Ratamess & French, 2002). However, it has been suggested that continuous bouts of intense RT are concomitantly associated with the accumulation of fatigue (Kataoka et al., 2022), although evidence is inconclusive on the topic. Deloads, characterized by short periods (∼1 week) of decreased training volume, load and/or intensity of effort, are a common strategy used by coaches and athletes to counteract accumulated fatigue and diminish the potential for nonfunctional overreaching (Bell et al., 2022). A recent study using the International Delphi Consensus technique defines deloads as “a period of reduced training stress designed to mitigate physiological and psychological fatigue, promote recovery, and enhance preparedness for the subsequent training cycle” (Bell et al., 2023); therefore, periods of complete training cessation, or detraining periods, could conceivably be considered one method by which deloads are employed to restore and rejuvenate. Although current research analyzing the effects of detraining is limited, multiple studies have demonstrated mechanistic and pragmatic benefits when deloads are implemented into a training program (Houmard et al., 1994; Ogasawara et al., 2013). Alternatively, these findings contrast with those of Vann et al. (2021), which may be explained by the length of the detraining periods used.

Some have speculated that the diminished rate of muscular adaptations typically seen in the latter phases of RT programs may also be negated with the implementation of detraining periods (Ogasawara et al., 2013). Indeed, short periods of cessation of training may attenuate the reduction in anabolic signaling protein phosphorylation typically seen with continuous bouts of RT (Jacko et al., 2022), as well as upregulate genes associated with muscle hypertrophy (Seaborne et al., 2018), facilitating a “re-sensitization” of muscle to hypertrophic stimuli; these findings suggest that cessation of training may be a particularly effective strategy during the deload period. Moreover, increases in serum testosterone and decreases in serum cortisol have been demonstrated following periods of detraining (Hortobágyi et al., 1993), which may potentiate (i.e., to enhance the effect of) muscular adaptations in following training cycle; this hypothesis remains speculative. Pragmatically, it has been demonstrated that the short-term reduction in volume load associated with deloads results in increased muscle size as well as increased performance in the barbell back squat (Hartmann et al., 2015; Ratamess et al., 2003).

Although the findings presented above are intriguing, current research on the effects of detraining does not reflect the typical practices of those in the lifting community (Bell et al., 2022). For instance, the length of detraining periods in the literature (i.e., 3 weeks) (Ogasawara et al., 2012; Ogasawara et al., 2013) are typically much longer than what is commonly employed in real-world settings (e.g., 5–7 days) (Bell et al., 2022). Moreover, to our knowledge there is no empirical evidence analyzing the direct potentiating effects of deloads on subsequent training cycles in resistance-trained individuals. Given the paucity of research on the topic, the purpose of this study was to investigate the effects of deloading, implemented as a 1-week period of cessation from training at the midpoint of a 9-week RT program, on muscular adaptations in resistance-trained individuals. We hypothesized that deloading would result in superior muscular adaptations potentially via re-sensitization of muscle to anabolic stimuli.

Material and Methods

Participants

We recruited 50 male and female volunteers from a university population. This sample size was justified by a priori precision analysis for the minimum detectable change at the 68% level (MDC68%; i.e., 1 standard deviation (SD), which is conservative in that it requires a larger sample to produce a narrow interval) for mid-thigh hypertrophy (i.e., SEM×2=2.93 mm), such that the compatibility interval (CI) of the between-group effect would be approximately ± MDC68%. Based on data from previous research (Schoenfeld et al., 2019), along with their sampling distributions, Monte Carlo simulation was used to generate 90% CI widths for 5,000 random samples of each sample size. To ensure a conservative estimate, as literature values may not be extrapolatable, the sum of each simulated sample size’s 90% CI’s mean and SD was used, and the smallest sample that exceeded MDC68% was chosen; that is, 18 participants per group (1:1 allocation ratio). Additional participants were recruited to account for the possibility of dropout. To incentivize participation and adherence, participants received monetary compensation for completing the study.

To qualify for inclusion in the study, the participants were required to be: (a) between the ages of 18–40 years; (b) free from existing cardiorespiratory or musculoskeletal disorders; (c) self-reported as free from consumption of anabolic steroids or any other illegal agents known to increase muscle size currently and for the previous year; and, (d) considered as resistance-trained, defined as consistently lifting weights at least 3 times per week (on most weeks) with at least 1 weekly session for the lower body muscles for at least 1 year. Participants were asked to refrain from the use of creatine products throughout the course of the study period, as this supplement has been shown to enhance muscle-building when combined with RT (Kreider et al., 2017).

Participants were randomly assigned to 1 of 2 experimental, parallel groups: An experimental group that deloaded (i.e., no RT) during the fifth week of a 9-week RT program (DELOAD: n = 25) or a traditional training group that performed the same RT program continuously over the study period (TRAD: n = 25). Randomization into groups was carried out using block randomization, with 2 participants per block, via online software (http://www.randomizer.org.). Approval for the study was obtained from the Lehman College Institutional Review Board (#2022-0762-Lehman). Written informed consent and completion of the 2022 PAR-Q+ were obtained from all participants prior to enrollment in the study. The methods for this study were preregistered prior to recruitment (https://osf.io/bztka). The supplemental files are available at: https://osf.io/kdgv3/. Portions of this text were previously published as part of a preprint (https://sportrxiv.org/index.php/server/preprint/view/302).

Resistance training procedures

The RT program was structured as an upper body/lower body split routine, with each body region protocol performed twice weekly. As previously described (Plotkin et al., 2022), the lower body protocol was directly supervised by the research team with each participant trained by at least one research assistant to monitor the proper performance of the respective routines and ensure participant safety. The research team consisted of over ten individuals, all with different training certifications ranging from multiple personal training certifications to none of any kind; everyone on the research team had a degree in an exercise-related field.

Exercises consisted of the Smith squat, leg extension, straight-leg toe press, and seated calf raise, in whichever order was available upon arriving to the lab. Participants performed 5 sets of 8–12 repetition maximum (RM) for each exercise with 2 min rest between sets. To help standardize the intensity of effort of the training protocols, we verbally encouraged participants to perform all sets to the point of volitional failure, herein defined as the inability to perform another concentric repetition while maintaining proper form. The cadence of repetitions was carried out in a controlled fashion, with a concentric action of approximately 1 s and an eccentric action of approximately 2 s as estimated by the research staff (i.e., without the use of a metronome). Loads were progressively adjusted from set to set within each session as well as across the duration of the study period to maintain the target repetition range. To enhance ecological validity, participants were given a mandatory upper body RT program to follow on alternate training days (without supervision by the researchers) and were instructed to refrain from performing any additional lower body RT for the duration of the study. Participants performed the upper body workouts at the time and location of their choosing, including the university’s fitness center, which all participants could access freely. Resources for 4x/week supervised training were not available, however, to enhance accountability, participants kept a training log of their upper body routines and emailed the log to the lead researcher on a weekly basis. Upper body workouts lasted approximately one hour. An overview of the training program is presented in File S1.

Prior to initiating the training program, participants underwent 10RM testing to determine individual initial loads for each lower body exercise. The RM testing was consistent with recognized guidelines as established by the National Strength and Conditioning Association (Baechle & Earle, 2008). Thereafter, training for both routines consisted of 4 (2 supervised, 2 unsupervised) weekly sessions performed on non-consecutive days for 9 weeks at whatever time was convenient for the participants between 9:00 AM and 4:00 PM. The DELOAD group took a 1-week break from training after the fourth week while the TRAD group trained consistently throughout the study period. The DELOAD group was instructed to refrain from resistance training of any kind during the fifth week, but were allowed to continue with aerobic and/or sport specific training. Participants were allotted two nonconsecutive missed sessions and were removed if they missed an entire week of training outside of the allowed deloading week for those in the DELOAD group.

Dietary adherence

To avoid potential dietary confounding of results, participants were advised to maintain their customary nutritional regimen as previously described (Plotkin et al., 2022). Dietary adherence was assessed by self-reported 5-day food records (including at least 1 weekend day) using MyFitnessPal.com (http://www.myfitnesspal.com), which has good relative validity for tracking energy and macronutrient intake (Teixeira et al., 2018). Nutritional data was collected twice during the study: 1 week before the first training session (i.e., baseline) and during the final week of the training protocol. Participants were instructed on how to properly record all food items and their respective portion sizes consumed for the designated period of interest. Each item of food was individually entered into the program, and the program provided relevant information as to total energy consumption, as well as the amount of energy derived from proteins, fats, and carbohydrates for each time-period analyzed.

Measurements

The following measurements were conducted pre- and post-study in testing sessions separated from the training sessions by at least 48 h. All measurements were taken in the same testing session, in the order that they appear in this manuscript, aside from the readiness to train questionnaire, which was provided 24–48 h after the final training sessions of weeks four and nine. Participants reported to the lab at the time of their choosing between 10:00 AM and 2:00 PM, having refrained from any strenuous exercise for at least 48 h prior to baseline testing and at least 48 h prior to testing at the conclusion of the study. Anthropometric and muscle thickness (MT) assessments were performed first in the session, followed by measures of muscle strength. Each strength assessment was separated by a 10-minute recovery interval to ensure restoration of resources.

Anthropometry: To reduce the potential for confounding from lifestyle factors, participants were told to refrain from eating or drinking for 8 h prior to testing, eliminate alcohol consumption for 24 h, and void their bladder immediately before anthropometric testing. Caffeine intake was not assessed, but the restriction on fluid consumption precluded intake of caffeinated beverages. Participants’ heights were measured using a stadiometer and assessments of body mass and percent body fat and segmental lower limb lean mass were obtained by multifrequency bioelectrical impedance analysis (Model 770, InBody Corporation, Seoul, South Korea) as per the instructions of the manufacturer.

Muscle Thickness: As previously described (Plotkin et al., 2022), ultrasound imaging was used to obtain measurements of MT. A trained ultrasound technician performed all testing using a B-mode ultrasound imaging unit (Model E1, SonoScape, Corporation, Shenzhen, China). The technician applied a water-soluble transmission gel (Aquasonic 100 Ultrasound Transmission gel, Parker Laboratories Inc., Fairfield, NJ) to each measurement site, and a 4–12 MHz linear array ultrasound probe was placed perpendicular to the tissue interface without depressing the skin. When the quality of the image was deemed to be satisfactory, the same technician saved the image to a hard drive and immediately obtained MT dimensions by measuring the distance from the subcutaneous adipose tissue-muscle interface to either the aponeurosis or the muscle-bone interface. The following measurements were conducted using identical procedures in pre- and post-study testing sessions. Measurements were taken on the right side of the body at the mid-thigh (a composite of the rectus femoris and vastus intermedius), lateral thigh (a composite of the vastus lateralis and vastus intermedius), medial gastrocnemius, lateral gastrocnemius, and lateral soleus muscles. For the quadriceps, subjects reclined in a supine position and measurements were obtained at 30%, 50% and 70% between the lateral condyle of the femur and greater trochanter. For the calf muscles, subjects assumed a prone position and measurements were taken on the posterior surface of both legs at 25% of the lower leg length (the distance from the articular cleft between the femur and tibia condyles to the lateral malleolus). To ensure that swelling in the muscles from training did not obscure MT results, images were obtained at least 48 h after exercise/training sessions both in the pre- and post-study assessment. This is consistent with research showing that acute increases in MT return to baseline within 48 h following a RT session (Barakat et al., 2019; Ogasawara et al., 2012) and that muscle damage is minimal after repeated exposure to the same exercise stimulus over time (Damas et al., 2016; Biazon et al., 2019). To further ensure accuracy of measurements, 3 successive images were obtained for each site and then averaged to obtain a final value.

Lower Body Muscle Power: Lower body muscle power was assessed via the vertical jump test. As previously described (Plotkin et al., 2022), each participant was instructed on proper performance of the countermovement jump (CMJ) prior to testing by one of two researchers (MC or RB). Performance was carried out as follows: The participant began by assuming a shoulder-width stance with the body upright and hands on hips. When ready for the movement, the participant descended into a semi-squat position and then forcefully reversed direction, jumping as high as possible before landing with both feet on the ground.

Assessment of jump performance was carried out using a Just Jump mat (Probotics, Huntsville, AL), which was attached to a hand-held computer that records airtime and thereby ascertains the jump height. The participant stood on the mat and performed 3 maximal-effort CMJs with a 1-minute rest period between each trial. Participants were provided feedback regarding their performance between jumps. The highest jump was recorded as the final value.

Isometric Muscle Strength: As previously described (Vigotsky et al., 2019), isometric strength assessment was carried out using dynamometry testing (Biodex System 4; Biodex Medical Systems, Inc. Shirley, NY, USA). After familiarization with the dynamometer and protocol, the participant was seated in the chair and performed unilateral isometric actions of the knee extensors on his/her dominant limb.

During each trial, the participant sat with his/her back flush against the seat back pad and maintained a hip joint angle of 85 degrees with the center of his/her lateral femoral condyle aligned with the axis of rotation of the dynamometer. The dynamometer arm length was adjusted to allow the shin pad to be secured with straps proximal to the medial malleoli. A strap was secured across the participant’s ipsilateral thigh, hips, and torso to help prevent extraneous movement during performance and the participant was instructed to hold onto handles for greater stability. Testing was carried out at a knee joint angle of 70-degrees (Knapik et al., 1983).

Each maximum voluntary contraction trial lasted 5 s and was followed by a 30-second rest period, for a total of 4 trials. Participants were verbally encouraged to produce maximal force throughout each contraction; however, we did not provide augmented feedback to participants during the assessment. The highest peak net extension moment from the four trials was used for analysis.

Dynamic Muscle Strength: Dynamic lower body strength was assessed by 1RM testing in the back squat (1RMSQUAT) exercise performed on the same Smith machine (Hammer Strength Equipment, Life Fitness, Rosemont, IL, USA) for all participants. As previously described (Plotkin et al., 2022), participants reported to the lab having refrained from any exercise other than activities of daily living for at least 48 h prior to baseline testing and at least 48 h prior to testing at the conclusion of the study. The RM testing was consistent with recognized guidelines as established by the National Strength and Conditioning Association (Baechle & Earle, 2008). In brief, participants performed a general warm-up prior to testing consisting of light cardiovascular exercise lasting approximately 5–10 min. Next, a specific warm-up set of the squat of 5 repetitions was performed at ∼50% 1RM followed by 1 or 2 sets of 2–3 repetitions at a load corresponding to ∼60–80% 1RM. Participants then performed sets of 1 repetition of increasing weight for 1RM determination, with a minimum increase of 2.3 kg between attempts. Three to 5 min rest was provided between each successive attempt, based on the participants’ subjective feeling of readiness between attempts. Participants’ upper thighs had to reach parallel in the 1RMSQUAT for the attempt to be considered successful. Confirmation of squat depth was obtained by a research assistant positioned laterally to the participant to ensure accuracy. 1RM determinations were made within 5 attempts.

Local Muscular Endurance: Absolute lower-body local muscular -endurance was assessed by performing the leg extension exercise on the same selectorized machine (Life Fitness, Westport, CT) for all participants using 60% of the participant’s initial body mass. The smallest possible incremental increase in load for the unit was ∼1.1 kg. As previously described (Plotkin et al., 2022), participants sat with their back flat against the backrest, grasping the handles of the unit for support. The backrest was adjusted so that the anatomical axis of the participant’s knee joint aligned with the axis of the unit. Participants placed their shins against the pad attached to the machine’s lever arm. Participants performed as many repetitions as possible using a full range of motion (90-0 degrees of knee flexion) while maintaining a constant cadence of 1 − 0 − 1 − 0 as monitored by a metronome (i.e., is 1 s concentrically, no pause at full extension, 1 s eccentrically, and no pause at full flexion). The test was terminated when the participant could not perform a complete repetition with proper form in tempo. Local muscular endurance testing was carried out after assessment of muscular strength to minimize effects of metabolic stress potentially interfering with performance of the latter.

Readiness to Train Questionnaire: To assess participants’ subjective feelings toward training across the study period, we employed a readiness-to-train questionnaire as previously described in the literature (Pedersen et al., 2022). The questionnaire comprised seven questions using Likert-type scales ranging from 1 to 4, 1 to 5 and 1 to 10 (see File S2). As previously explained (Pedersen et al., 2022), the upper and lower boundaries of the scale were defined as follows: “1 can be described as not at all/extremely low and 4, 5, 10 (depending on lower/upper end of the scale) can be described as extreme amount/extremely high”. The questionnaire was given to participants 24–48 h after the fourth and ninth weeks of the study.

Blinding

To minimize the potential for bias, both the sonographer who conducted ultrasound testing and the statistician who analyzed data were blinded to group allocation.

Statistical analyses

All analyses were conducted in R (version 4.2.0) (R Core Team, 2019) within a Bayesian framework, with descriptive values expressed in means  ± SDs. Bayesian statistics represents an approach to data analysis and parameter estimation based on Bayes’ theorem (Van de Schoot et al., 2021) and can provide several advantages over frequentist approaches including: (1) formal inclusion of information regarding likely differences between interventions based on knowledge from previous studies (i.e., through informative priors); (2) flexible model building to capture a range of complexities within the data; and (3) presentation of inferences based on intuitive probabilities (Kruschke & Liddell, 2018; Van de Schoot et al., 2021). Inferences were not drawn on baseline nor within-group change, as baseline testing is inconsequential (Senn, 1994) and within-group outcomes are not the subject of our research question (Bland & Altman, 2011), although we descriptively presented within-group changes to help contextualize our findings. The effects of group (DELOAD vs. TRAD) on outcome variables were estimated using univariate and multivariate multilevel regression models (Vickerstaff, Ambler & Omar, 2021). Use of multivariate models improves precision by modeling all outcome variables simultaneously, taking advantage of the correlations between outcomes (Vickerstaff, Ambler & Omar, 2021) and avoiding limitations associated with separate inferences with related outcomes (Rubin, 2021). Additionally, the multilevel component of the analysis accounted for the repeated measures made on each participant across outcomes and time points. Recent data quantifying comparative distributions and correlations across outcomes following interventions in strength and conditioning were used to obtain informative priors (Swinton & Murphy, 2022). Inferences were made based on estimates of the difference in change between DELOAD and TRAD and their credible intervals.

Secondary analyses were performed on nutrition and readiness to train data, which were analyzed using multilevel regression models. Individual Likert readiness to train items were summed to create scales suitable for linear models assuming normal distribution of errors. All analyses were performed using the R wrapper package brms interfaced with Stan to perform sampling (Burkner, 2017). There are three main areas where Bayesian analyses can be performed inappropriately and/or result in poor inferences. These areas include: (1) issues related to prior selection; (2) misinterpretation of Bayesian features and results; and (3) improper reporting (Depaoli & Van de Schoot, 2017). To improve accuracy, transparency and replication in the analyses, the WAMBS-checklist (When to worry and how to Avoid Misuse of Bayesian Statistics) was used and we incorporated sensitivity analyses of influential data points and priors, which has been shown to be important in all cases including when diffuse priors are used (Depaoli, Winter & Visser, 2020). As identified in more detail in the File (S3), prior distributions for analyses presented in text included normal distributions. For the intercept, the mean and standard deviation were calculated using data from previous interventions in strength and conditioning and scaled relative to the baseline standard deviation (Swinton & Murphy, 2022). For the group difference, the mean was set to zero and standard deviation calculated to represent comparative differences expected in strength and conditioning (Swinton & Murphy, 2022). To assess bias following different variance specification, gamma distributions were used with the scale parameter set to 1, and the shape parameter ranging from 1 to 35 depending on the outcome (File S3).

Results

Of the initial 50 participants who volunteered to participate, 39 completed the study (DELOAD: n = 18 (12 male, six female), height (cm) = 170.7 ± 7.7, weight (kg) = 77.7 ± 15.8, age (yrs) = 22.2 ± 6.1, training experience (yrs) =  3.7  ± 4.5; TRAD: n = 21 (17 male, 4 female), height (cms) = 172.9 ± 8.8, weight (kg) = 79.1 ± 13.5, age (yrs) = 21.4 ± 3.9, training experience (yrs) = 3.2  ± 2.6). Reasons for dropouts were: Personal reasons (n = 5), lack of compliance (n = 5), and training-related injury not related to the study (n = 1). All participants that completed the study attended >85% of the total sessions, with both groups displaying an average attendance of ∼96%. Figure 1 displays a CONSORT diagram of the data collection process. Table 1 presents a descriptive summary of the pre- and post-intervention values for all outcomes.

Figure 1 Consort diagram.

CONSORT flow chart of the data collection process.

Body composition and muscle morphology

Initial univariate analyses are presented in Table 2. The evidence obtained did not support greater body composition changes when including a period of deloading as indicated by median group difference estimates close to zero, and all 95% credible intervals substantially overlapping zero. Posterior probabilities that group differences favored the inclusion of a period of deloading were generally low (0.273 ≤ p ≤ 0.835; Table 1). Multivariate analysis comprising muscle thickness measurements did not alter findings (Table 2). Illustration with standardized mean difference effect sizes showed consistency in results and that if group differences did exist, they were likely to be small in magnitude (Fig. 2). Calculation of within group differences demonstrated that both groups achieved positive adaptations with small to medium increases in muscle thickness; however, body fat percentage and lower body lean mass showed minimal change (see File S3). Diagnostic evaluations across all analyses identified no causes for concern and no changes in conclusions based on sensitivity analyses (see File S3).

Strength and performance

Initial univariate analyses are presented in Table 3. Results were inconsistent, with median group difference estimates close to zero and 95% credible intervals substantially overlapping zero for endurance and CMJ performance (Table 3). In contrast, some evidence was obtained for greater isometric and dynamic strength adaptations of TRAD relative to inclusion of a deloading period (Table 3), with posterior probabilities that group differences favored TRAD equal to p = 0.851 for 1RM, and p = 0.924 for isometric strength. Multivariate analysis for strength outcomes did not alter findings (Table 3). Illustration with standardized mean difference effect sizes showed that if group differences did exist, they were likely to be small in magnitude for endurance and CMJ performance (Fig. 3), whereas they may be small to large in favor of TRAD for 1RM and isometric strength. Calculation of within group differences were mixed with some evidence that both groups improved across all variables (see File S3). Diagnostic evaluations across all analyses (see File S3) identified no causes for concern, with sensitivity analyses producing similar findings to those presented in the main text.

Table 1 Descriptive summary of pre- and post-intervention values for all outcomes.

	DELOAD (n = 18)	TRAD (n = 21)	
Variable	Pre	Post	Pre	Post	
1RM (kg)	92.8 ± 38.5	105.8 ± 32.1	95.9 ± 21.7	112.3 ± 21.3	
Isometric Strength (N ⋅m)	258.8 ± 60.6	261.8 ± 70.5	268.4 ± 55.0	288.6 ± 55.0	
Mid-quad 30% (mm)	50.8 ± 8.3	54.3 ± 8.8	53.6 ± 8.2	57.1 ± 8.0	
Mid-quad 50% (mm)	41.4 ± 8.1	45.5 ± 9.0	44.7 ± 8.1	49.3 ± 7.5	
Mid-quad 70% (mm)	29.8 ± 7.0	33.9 ± 8.0	32.1 ± 6.4	36.0 ± 6.5	
Lateral quad 30% (mm)	34.2 ± 5.9	36.5 ± 6.0	34.2 ± 7.9	36.6 ± 7.8	
Lateral quad 50% (mm)	36.0 ± 5.4	38.8 ± 5.7	36.6 ± 6.5	39.6 ± 6.8	
Lateral quad 70% (mm)	31.5 ± 4.8	34.4 ± 5.3	32.7 ± 4.9	34.9 ± 5.6	
Medial Gastrocnemius (mm)	19.3 ± 4.2	20.5 ± 3.7	19.2 ± 2.7	20.6 ± 2.8	
Lateral Gastrocnemius (mm)	16.5 ± 2.5	17.3 ± 2.4	16.5 ± 3.5	17.6 ± 3.5	
Soleus (mm)	15.2 ± 3.2	16.2 ± 3.8	15.7 ± 3.3	16.3 ± 3.4	
Counter Movement Jump (cm)	39.9 ± 9.4	41.4 ± 9.1	45.2 ± 8.4	46.0 ± 9.7	
Strength Endurance (reps)	16.3 ± 6.0	20.4 ± 3.8	15.5 ± 5.8	20.6 ± 6.9	

Table 2 Multivariate and univariate analyses of potential group pre to post differences for body composition and muscle growth outcomes.

Variable	Multivariate Group Difference [95%CrI]	Posterior probability favoring inclusion of detraining	Univariate Group Difference [95%CrI]	Posterior probability favoring inclusion of detraining	
Rectus femoris 30% (mm)	−0.33 [−2.0 to 1.4]	p = 0.347	−0.16 [−2.1 to 1.8]	p = 0.434	
Rectus femoris 50% (mm)			−0.63 [−2.8 to 1.5]	p = 0.273	
Rectus femoris 70% (mm)	−0.17 [−1.9 to 1.6]	p = 0.563	
Vastus lateralis 30% (mm)	0.08 [−1.5 to 1.6]	p = 0.540	−0.07 [−1.8 to 1.7]	p = 0.466	
Vastus lateralis 50% (mm)			−0.27 [−1.9 to 1.4]	p = 0.373	
Vastus lateralis 70% (mm)	0.53 [−1.2 to 2.2]	p = 0.730	
Lateral gastrocnemius (mm)	−0.07 [−0.65 to 0.48]	p = 0.400	−0.23 [−1.2 to 0.71]	p = 0.317	
Medial gastrocnemius (mm)			−0.22 [−1.0 to 0.59]	p = 0.290	
Soleus (mm)	0.35 [−0.36 to 1.0]	p = 0.835	
Body fat (%)	*	*	−0.10 [−1.2 to 1.1]	p = 0.424	
Lower body lean mass (kg)	*	*	−0.12 [−0.37 to 0.14]	p = 0.185	
Notes.

Multivariate analysis of muscle thickness data combined for single rectus femoris, vastus lateralis, and calf thickness variables.

* Not included in analysis

Figure 2 Muscle morphology.

Posterior distributions of group differences for body composition and muscle morphological outcomes expressed as standardized mean difference effect sizes. Negative values favor control and positive values favor the inclusion of a detraining period. Effect sizes were calculated by dividing group differences by the pooled baseline SD. Small (0.15), medium (0.30) and large (0.50) thresholds derived for strength and conditioning interventions are presented with gray lines.

Table 3 Multivariate and univariate analyses of potential group pre to post differences for performance variables.

Variable	Univariate Group Difference [95%CrI]	Posterior probability favoring inclusion of detraining	Univariate Group Difference [95%CrI]	Posterior probability favoring inclusion of detraining	
Isometric (N ⋅m)	−11.5 [−33.5 to 8.2]	p = 0.245	−14.4 [−34.3 to 5.8]	p = 0.076	
One-repetition maximum (kg)	−4.5 [−10.4 to 2.8]	p = 0.116	−3.6 [−10.4 to 3.2]	p = 0.149	
Local Muscular Endurance (repetitions)	*	*	−0.55 [−2.9 to 1.9]	p = 0.321	
Countermovement jump (cms)	*	*	0.61 [−1.5 to 2.8]	p = 0.715	
Notes.

* Not included in analysis.

Figure 3 Strength performance.

Posterior distributions of group differences for performance outcomes expressed as standardized mean difference effect sizes. Negative values favor control and positive values favor the inclusion of a detraining period. Effect sizes were calculated by dividing group differences by the pooled baseline SD. Small (0.15), medium (0.30) and large (0.50) thresholds derived for strength and conditioning interventions are presented with gray lines.

Secondary analyses

Results from secondary analyses are presented in the supplementary file. No substantial evidence was found to indicate a difference in nutritional intake between groups. Some evidence was obtained to indicate greater sleep quality in the deload group at mid-intervention, and greater muscle soreness in the deload group at post-intervention with and without adjusting for mid-intervention values (File S3).

Discussion

This is the first study to directly assess the potentiating effects (i.e., potential to enhance the efficacy) of a 1-week deload period on muscular adaptations. Our novel results suggest that a 1-week deload, in the form of complete cessation from training, has a minimal impact on measures of muscle hypertrophy, endurance, or power in the context of a 9-week training block; correspondingly, we found no evidence of a potentiating effect pursuant to re-sensitization. Conversely, while both groups increased strength, TRAD experienced modest benefits in measures of both isometric and dynamic strength. In the ensuing sections, we discuss these results within the context of the current literature as well as their practical implications for exercise prescription.

Hypertrophy

Both groups increased muscle size over the course of the study, with similar between-group increases observed in all measurements. These findings suggest that 1 week of deloading, carried out as a cessation of training, does not attenuate the hypertrophic adaptations seen in the first half of a 9-week training block but also does not enhance results over time. The findings are generally consistent with the body of literature, which suggests little to no differences in longitudinal muscle growth when relatively short periods of training cessation are utilized (Ogasawara et al., 2011; Ogasawara et al., 2013). Previous studies on the topic employed longer periods of cessation of training (3 weeks), recruited untrained participants, and used relatively low-volume RT protocols (9 total sets/muscle group/week) specific to the bench press exercise (Ogasawara et al., 2011; Ogasawara et al., 2013), thus compromising the ecological validity of findings. Alternatively, the design of our investigation aligns more closely with the manner in which deloads are commonly employed by coaches and athletes in the field, thus filling an important gap in the literature (Bell et al., 2022).

We originally hypothesized that individuals in DELOAD would experience superior muscle growth due to the dissipation of fatigue accrued in the first 4 weeks of training and potential re-sensitization to hypertrophic stimuli. However, although no objective measures of fatigue or anabolic signaling were assessed, participants anecdotally often reported feeling lethargic (i.e., out of practice) after the deloading period rather than refreshed. This corroborates the findings by Hortobágyi et al. (1993), and although speculative may be explained by the fact that participants in the deload group did not train during the fifth week, rather than using deload paradigms often employed by coaches and athletes in strength and physique sports that involve reduced training volumes and/or intensities (Bell et al., 2022; Hortobágyi et al., 1993). Perhaps a period of reduced training volume and intensity, but not complete cessation, would allow for the dissipation of fatigue without bringing about a feeling of lethargy upon return. Whether different deload paradigms may result in hypertrophic benefits warrants further investigation.

Strength

Both groups experienced increases in dynamic and isometric strength; however, these measures generally showed superiority for TRAD. The between-group differences were most apparent in the isometric knee extension, where the CIs encapsulated effects ranging from a small negative effect to a large positive effect favoring TRAD (−5.1 and 42.1 nM, respectively). For 1RM squat testing, the results were somewhat more equivocal, but nevertheless indicate a potential benefit for TRAD. The spread of the CIs encapsulated effects ranging from a modest negative effect to an appreciable positive effect favoring TRAD (−3.0 and 12.1 kg, respectively).

The relative benefits seen by those in the TRAD group are unexpected given that the current body of literature suggests relatively short periods of training cessation have little to no effect on strength (Ogasawara et al., 2011; Ogasawara et al., 2013). However, it is important to note that the multiple instances of 1RM testing used by Ogasawara et al. may explain these discrepancies (Ogasawara et al., 2011; Ogasawara et al., 2013). These findings are particularly surprising considering the extensive use of deloads in athletes involved in strength sports (i.e., powerlifting and weightlifting) (Bell et al., 2022). It is important to note that the aim of RT protocol in this study was not to maximize strength, but rather to maximize hypertrophy (i.e., moderate loads, higher volumes). Therefore, it is conceivable that deloads may confer different effects when employing an RT protocol consistent with that of strength athletes (i.e., the use of higher percentages of 1RM). It also is unknown if a brief period of reduced training (i.e., not total training cessation), similar to deload strategies often employed in the field, may help to attenuate the observed blunting of strength gains or perhaps even potentiate improvements. These hypotheses should be explored in future research.

Another variable that warrants consideration is that of specificity. Although both strength assessments suggested superior improvements for TRAD, isometric outcomes showed a greater benefit than dynamic testing. Although speculative, it is conceivable that this discrepancy may be attributed to the specificity of transfer between use of Smith machine squats in both the training and testing protocols. Simply stated, the 1-week deload period may have had a true negative impact on strength, but the similarities between the training and dynamic testing somewhat masked those detriments, whereas the lack of transfer from training to isometric testing did not. This hypothesis warrants further investigation.

Local muscular endurance

Leg extension endurance slightly favored the DELOAD group. However, the magnitude of difference between groups was less than a single repetition, thus not likely to be of practical significance. Research regarding the potentiating effects of deloading on local muscle endurance is very limited, making it difficult to compare our results with similar study designs (Coratella & Schena, 2016; Sysler & Stull, 1970).

It has been proposed that local muscular endurance performance is predicated on adaptations including increases in capillarization and mitochondria activity as well as enhanced metabolic enzymatic activity (Haff & Triplett, 2015). Interestingly, all these adaptations seem to be negatively impacted by short periods of complete training cessation (Mujika & Padilla, 2001). Additionally, increases in maximal strength have been speculated to enhance local muscular endurance due to loads used in testing being a lower percentage of an individual’s 1RM, though evidence is inconclusive on the topic (Schoenfeld et al., 2021). Therefore, it is possible that periods of deloading may further hinder muscular endurance adaptations because of their concomitant detriments to maximal strength. However, this did not appear to occur with the deload period employed in our study. Moreover, we did not assess 1RM strength in the leg extension and therefore it is not clear whether increases in dynamic strength could have played a role in results (Chatlaong et al., 2022).

A similar issue to strength data extrapolation can be seen in our muscle endurance results. Specifically, this study design employed a moderate repetition range (8–12 repetitions), whereas muscle endurance is seemingly best trained through sets containing 15 or more repetitions (Schoenfeld et al., 2021). Thus, it is possible that training with the explicit goal to elicit increases in muscular endurance may yield alternate results, although recent meta-analytic work challenges this hypothesis (Hackett et al., 2022). More research is needed to fully understand the effects of deloading on local muscular endurance.

Muscular power

Differences between groups in CMJ performance were trivial. Our findings are generally consistent with the body of literature, which suggests power adaptations observed in training are not attenuated by short periods (<2 weeks) of detraining (Hortobágyi et al., 1993). Importantly, our protocol required that participants control each repetition both eccentrically and concentrically, likely resulting in little adaptation to the stretch shortening cycle used in explosive movements. Perhaps greater differences between groups would be realized by incorporating plyometric-based training into the design (Griffiths et al., 2019). Whether different RT designs will result in differences in lower body power following deloading warrants further investigation.

Readiness to train

Participants in the TRAD group showed potential advantage in their perception of some readiness to train components compared to those in the DELOAD group. For example, the DELOAD group reported an increase in muscle soreness whereas individuals in the TRAD group reported decreases in soreness from week 4 to week 9. Additionally, individuals in the DELOAD group reported a decrease in motivation to train from week 4 to 9 as opposed to those in the TRAD group, who reported no differences in motivation. The magnitude of differences in these values can be considered relatively modest and their practical meaningfulness thus remains questionable.

In an attempt to promote functional overreaching (i.e., a supercompensation of fitness characteristics following short periods of training that exceed a systems capacity to recover), we employed a relatively high-volume program. Additionally, the participants were pushed to volitional failure on each set during the supervised aspect of the protocol and instructed to do the same during unsupervised upper body training. In total, the participants performed 90 weekly sets for all muscle groups combined during each training week of the intervention period. On the final testing day, participants were asked if they felt the need for a deload following the study period. During these post-study conversations, virtually every participant stated that they trained consistently harder than at any point in their previous training experience. However, quite surprisingly, almost none of the participants felt they needed a break after the study, with nearly all stating they would return to normal training routine within a couple of days of the study’s completion. Therefore, our findings suggest that achieving an overreaching or overtraining state from RT alone is unlikely, at least over relatively short training periods with ecologically valid protocols, which is consistent with current evidence on the topic (Grandou et al., 2020; Kataoka et al., 2022).

The present findings warrant speculation as to the possible use of autoregulatory deloads versus more proactive deloads. Our results suggest that, from a strength-related standpoint, having participants perform a deload even if they do not feel the need for a break may do more harm than good. This is perhaps why more strength and physique coaches prefer to employ a flexible deload approach as opposed to a more pre-planned paradigm (Bell et al., 2022). Whether the use of an autoregulated deload would result in differential results warrants further investigation.

Limitations

Our study contained multiple limitations that should be noted when extrapolating the findings to ecologically valid settings. First and foremost, this experiment was conducted on young men and women with a minimum of 1 year training experience. Therefore, our findings cannot necessarily be generalized to other populations including individuals over the age of 40, adolescents, and untrained individuals. Second, participants were not required to have training experience specific to the Smith machine squat. Thus, increases in 1RM strength may have been influenced by neural adaptations that would not likely be seen by individuals who regularly perform variations in the Smith machine back squat in their training program. Third, while research assistants verbally encouraged participants to perform sets with maximum intensity of effort, some individuals volitionally ended their sets prior to reaching momentary muscular failure throughout the study period. However, all participants trained with a high level of effort on all supervised sets; thus, any differences in proximity to failure likely had little consequence on study outcomes. Fourth, the outcomes assessed in this study were specific to the lower body musculature; thus, inferences regarding the effect of deloading on the upper body muscles cannot be drawn. To this point, while we can be confident that all participants trained with high intensities of effort during the supervised lower body sessions, we cannot be sure as to the effort exerted during upper body training. Although we attempted to collect weekly upper body training logs from each participant as to their upper body routines, the quality of reporting was often inconsistent, thus raising uncertainty about overall adherence to this aspect of the program. Fifth, we employed a pre-planned deload after a 4-week training cycle, which is a common strategy employed in real-world settings. However, we cannot necessarily draw inferences as to the effect of deloads after longer training cycles or autoregulated deloads on muscular adaptations. Sixth, our findings are the result of a short, 9-week training block and a high training volume (90 weekly sets) and relatively low frequencies (i.e., each muscle trained only twice weekly). Therefore, questions remain regarding the effects of deload periods within the context of longer training periods as well as higher weekly training volumes and frequencies. Seventh, markers of anabolic signaling were not measured, precluding us from drawing direct insights to the potential re-sensitization effect of deloads. Eighth, a time-matched control would conceivably have helped to account for measurement error and biological variability. However, measurement error and biological variability are also reflected in the TRAD condition (which essentially served as a control), thus accounting for random fluctuations or time trends that are not of interest to the study purpose. Moreover, it would be infeasible to recruit a group of resistance-trained subjects to cease training for ∼10 weeks, which would preclude the ability to conduct studies in this population (Beato, 2022). Finally, our results are specific to a deload involving a cessation of RT. In practice, deloads can employ a wide range of strategies designed to reduce training load, volume and/or intensity as opposed to abstention. Future studies should seek to investigate the effects of different deload approaches on muscular adaptations.

Conclusion

The implementation of a 1-week deload period at the midpoint of a 9-week training block produced similar increases in lower body muscle size, endurance, and power when compared to a continuous training block. These results suggest that both continuous and periodic training blocks are viable options when attempting to maximize hypertrophy, at least within a 9-week period. Conversely, continuous training showed superior improvements in measures of lower body strength compared to deloading. Thus, when trying to optimize increases in maximal strength, periods of complete training cessation likely should be used more sparingly. Ultimately, more research is needed to fully elucidate when and how deloads can be employed to maximize muscular adaptations as well as to determine for which populations these periods are best suited. From a research standpoint, our results suggest that relatively short-term investigations (≤ 9 weeks) with training volumes ≤ 90 total sets per week do not require deloads to facilitate recovery in young participants. Future studies should endeavor to investigate deloads that employ more extreme training volumes over longer time periods to determine whether these variables influence results.

Supplemental Information

Supplemental Information 1 Training Program

Click here for additional data file.

Supplemental Information 2 Readiness-to-Train Questionnaire

Click here for additional data file.

Supplemental Information 3 WAMBS checklist

Click here for additional data file.

We are grateful to the following research assistants for their help in data collection: Leonardo Generoso, Max Sapuppo, Max Joo-Yeop Kim, Kurt Roderick, Elizabeth Arias and Adam Mohan.

Additional Information and Declarations

Competing Interests

Author Contributions

Human Ethics

Data Availability

Brad J Schoenfeld serves on the scientific advisory board of Tonal Corporation, a manufacturer of fitness equipment. Michael Israetel is a cofounder of Renaissaince Periodization. The other authors declare that they have no competing interests.

Max Coleman conceived and designed the experiments, performed the experiments, prepared figures and/or tables, authored or reviewed drafts of the article, and approved the final draft.

Ryan Burke performed the experiments, authored or reviewed drafts of the article, and approved the final draft.

Francesca Augustin performed the experiments, authored or reviewed drafts of the article, and approved the final draft.

Alec Piñero performed the experiments, authored or reviewed drafts of the article, and approved the final draft.

Jaime Maldonado performed the experiments, authored or reviewed drafts of the article, and approved the final draft.

James P. Fisher conceived and designed the experiments, authored or reviewed drafts of the article, and approved the final draft.

Michael Israetel conceived and designed the experiments, authored or reviewed drafts of the article, and approved the final draft.

Patroklos Androulakis Korakakis performed the experiments, authored or reviewed drafts of the article, and approved the final draft.

Paul Swinton conceived and designed the experiments, analyzed the data, prepared figures and/or tables, authored or reviewed drafts of the article, and approved the final draft.

Douglas Oberlin performed the experiments, authored or reviewed drafts of the article, and approved the final draft.

Brad J. Schoenfeld conceived and designed the experiments, analyzed the data, prepared figures and/or tables, authored or reviewed drafts of the article, and approved the final draft.

The following information was supplied relating to ethical approvals (i.e., approving body and any reference numbers):

The Lehman College IRB granted ethical approval to carry out the study within its facilities

The following information was supplied regarding data availability:

The data and supplementary material are available at Open Science Framework: Schoenfeld, Brad J. 2023. “The Effects of a One-Week Deload Period during Regimented Resistance Training on Muscular Adaptations.” OSF. December 21. doi:10.17605/OSF.IO/KDGV3.

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
