# Peer review of "Gaining more from doing less? The effects of a one-week deload period during supervised resistance training on muscular adaptations"

_PeerJ, doi:10.7717/peerj.16777_

## Round 0.1 · original submission · Major Revisions

While the two reviewers and I both see some merit in your study and the submitted manuscript, they've also provided a comprehensive critique that has highlighted a number of ways in which your manuscript can be improved. I suggest you look to address these comments and resubmit the manuscript when you feel it is ready.

Reviewer 1 ·

Excellent Review

This review has been rated excellent by staff (in the top 15% of reviews)
EDITOR COMMENT
The reviewer was very thorough and comprehensive in the review of this manuscript. They provided highly detailed and articulate comments that were specific and actionable by the authors. Some of this feedback reflected the reviewer's view that the authors may have not correctly interpreted all aspects of some studies they cited, with the reviewer providing a clear direction on what they felt the previous studies actually found. As a result of this constructive objective detailed feedback, I feel the revised manuscript will be much stronger.

Basic reporting

***PLEASE SEE TO WORD DOCUMENT FOR ALL OF MY COMMENTS***

*Lines 24-25: I very much understand that it is a long held belief that fatigue accumulates with resistance training. I am, however, not certain what is actually accumulating, nor do I believe the resistance training literature as a whole has been able to answer such a question. The Vieira et al. review and meta-analysis reference (cited by Coleman et al.) compared acute markers of fatigue between resistance exercise protocols performed to failure and not to failure. Among the main findings was that performing resistance exercise to failure resulted in greater reductions in biomechanical properties (e.g., isometric strength) and metabolic responses (e.g., lactate concentrations) acutely after resistance exercise, as well as markers and muscle damage (e.g., creatine kinase activity) up to 48 hours following exercise. I do not know where in the Vieira paper that it is mentioned that fatigue accumulates with continuous bouts of resistance exercise. The closest I can find is “inappropriate fatigue management may increase the risk of injury, overtraining, and psychological burnout.” I recommend that Coleman et al. read a paper from Kataoka et al. 2021 wherein the evidence for the suggestion that fatigue accumulates with resistance training is explored. Kataoka et al. conclude that evidence for presence of fatigue accumulation with resistance training is equivocal. A lack of evidence for X does not inherently provide evidence against X, but it certainly makes it difficult to state that fatigue “accumulates” from resistance training. I recommend Coleman et al. consider altering their introduction accordingly, and if not, then please inform the readers on what the term “fatigue accumulates” means, with appropriate citations.

Kataoka et al 2021: https://pubmed.ncbi.nlm.nih.gov/34613589/

*Lines 30-32: I recommend the authors cite the Bell et al. 2022 paper. I also recommend Coleman et al. alter their statements such that they are supported by the Vann et al. study, which compared a 1 week period of active recovery (i.e., reduction in training volume) and passive recovery (i.e., complete cessation from training) after a 6-week period of high volume resistance training.

*Lines 33-35: The word several is used, yet the authors include only three citations. I recommend the authors either add more references or remove the word “several.” The references provided by Coleman et al. are highlighted below with comments:

- Vann et al. 2021 examined the effects of active and passive recovery on a variety of outcome measures (e.g., body composition, muscle fibre CSA, serum markers, and expression of molecular targets in skeletal muscle) after a 6-week period of high volume resistance training. They did not include a group that continued training during the 7th week, which limits the ability to state that any mechanistic or pragmatic benefits can be attained from including a period of detraining. I am also not certain what mechanistic or pragmatic benefits the study of Vann et al. demonstrated. I recommend Coleman et al. please clarify.

- Houmard et al. 1994 examined different “tapering” strategies (defined therein primarily as reductions in training volume) on running performance and running economy. None of the groups actually ceased training, and I am not sure what relevance (if any) such outcome measures have for Coleman et al.’s study. I recommend Coleman et al. remove the Houmard et al. 1994 citation.

- Ogasawara et al. 2013 examined changes in muscle size and strength between a periodic training group (i.e., 6-weeks of bench press training, followed by 2 cycles of 6 weeks of training + 3 weeks detraining) and a continuous training group (i.e., 24-weeks of bench press training without any interruptions). The periodic training group performed 25% fewer total training sessions than the continuous training group, yet demonstrated similar adaptations compared to the continuous training group after the 24-week training period. I recommend that Coleman et al. highlight these findings in their introduction (i.e., in the “pragmatic portion of the introduction; please see below). I also recommend that Coleman et al. read another paper from the Ogasawara group (i.e., Ogasawara et al. 2013), wherein the researchers show that the attenuated responses of p70S6K and rpS6 phosphorylation after 12 and 18 bouts of exercise could be restored after a short period of detraining (i.e., 12 days) without any significant morphological changes to the muscle itself. I recommend Coleman et al. cite the Ogasawara et al. 2013 paper.

Ogasawara et al. 2013: https://pubmed.ncbi.nlm.nih.gov/23372143/

*Lines 35-38: The references provided by Coleman et al. are highlighted below with comments:

- Hortobágyi et al. 1993 indeed demonstrated significant increases and decreases, respectively, in serum testosterone and cortisol, but the study was a cross-sectional design. The Vann et al. 2021 study, which was an experimental design did not confirm these findings. The contrast in findings between studies could hypothetically be explained by differences in the durations of the detraining/deloading periods. I recommend that Coleman et al. at least highlight these inconsistencies, and note the time period for which the participants in Hortobágyi et al. refrained from their normal training (i.e., 14 days).

- The review paper Mujika and Padilla 2000 centers around physiological and performance outcomes as a consequence of detraining (therein defined as an insufficient training stimulus to maintain training induced adaptations). The work of Mujika and Padilla discusses cardiorespiratory (e.g., maximal oxygen uptake, heart rate) and metabolic detraining (e.g., muscle glycogen), as well as a little bit on muscle fibre characteristics and enzymatic activity, strength performance, and the hormonal responses to detraining. As far as I can tell, Mujika and Padilla do not mention anything regarding the ability of a detraining period to potentiate adaptations. They largely focus on endurance training adaptations, and reference the Hortobágyi et al. 1993 study, but do not state anything more than Hortobágyi’s results. I recommend that Coleman et al. remove the Mujika and Padilla citation and either alter, or find another work of literature to support their statements.

*Lines 38-41: Jacko et al. 2022 indeed provide evidence of a re-sensitization to anabolic signalling after 10 days of detraining, but I think there are inherent issues to their design (i.e., pooling data, secondary analyses, and as mentioned by Jacko et al., being statistically underpowered). I recommend that Coleman et al. include the Ogasawara et al. 2013 study provided above as a reference. I also recommend Coleman et al. read a paper by Damas et al. 2018. An interesting talking point, which Coleman et al. might consider highlighting in their introduction, is that it is largely unknown whether the re-sensitization of the muscle to anabolic signalling truly has any beneficial effects on muscle growth, or whether it is merely an expression of newly proteins due to increased structural damage.

Damas et al. 2018: https://pubmed.ncbi.nlm.nih.gov/29282529/

*Lines 42-44: The reference provided by Coleman et al. is highlighted below with comments:

- Ratamess et al. 2003 examined the effects of amino acid supplementation on upper and lower body strength, upper and lower body peak power, and “high-intensity” local muscular endurance. The amino acid supplementation and placebo groups both demonstrated increases in squat 1RM strength during the last week of the overtraining periods, and further increased 1RM squat after the two week period of reduced training volume (Figure 2). Bench press 1RM showed similar results for the placebo group, but the amino acid supplementation group did not further increase 1RM bench press (i.e., after the two week period of reduced volume training; Figure 1). The two groups did not demonstrate any significant increases in upper or lower body peak power in response to two weeks of reduced training volume (i.e., from the end of the overtraining period to the end of the reduced volume training period; Figure 3, Figure 4). I recommend Coleman et al. consider if Ratamess et al. 2003 is an appropriate citation given the use of supplementation, and for the following two reasons: 1) investigators did not include a group that continued to training without reduced volume, which limits the ability to state that any pragmatic benefits can be attained from including a period of detraining per se and 2) the impact of tapering (i.e., reducing training volume) on 1RM strength and power was less dramatic than 1RM strength changes observed during the actual training phase. If Coleman et al. still believe Ratamess et al. 2003 is an appropriate citation, then I recommend their statements be altered accordingly to reflect Ratamess’s data. I suggest that Coleman et al. remove the Ratamess et al. citation completely, and include the findings from the Ogasawara papers (i.e., the fact that the periodic and continuous training groups demonstrated similar adaptations after the 24 weeks training period notwithstanding the reduction in training sessions for the periodic training groups), as they arguably could be the most pragmatic reason to include a deload.

*Lines 44-46: The reference provided by Coleman et al. is highlighted below with comments:

- Ogasawara et al. 2011 found that 6-weeks of bench press training, followed by 3-weeks of detraining and then another 6-weeks of bench press training resulted in similar adaptations in muscle size and strength compared to 15-weeks of training without interruptions. These results suggest that a 3-week detraining period might not negatively influence or enhance muscular strength and size adaptations; however, I am not sure they can be used to support Coleman et al.’s statement the diminished rate of muscular adaptations typically seen in the latter phase of training can be negated with the inclusion of deloads. I am aware Ogasawara reported percent changes in muscle size across the 15-weeks of training (Table 2); however, I am not certain if the authors actually compared the percent changes between time points. I recommend Coleman read a paper by Counts et al. 2017 on the time course of muscle growth, wherein the authors discuss using percentages to calculate the percentage change in muscle size per day. Per Counts et al: “Although this may be meaningful for estimating overall muscle growth, this provided no information regarding when muscle growth occurred or whether there was a plateau in muscle growth.” I would also like Coleman et al. to reconsider whether the attenuated rates of muscle growth and strength are truly “negated” if, at the end of 15-weeks of training, Ogasawara et al. found that the increases in muscle size and strength were not different between the two training programs. Indeed, if anything, the continuous training program actually appears to be numerically higher in terms of total improvement percent change in triceps brachii and pectoralis major muscle CSAs and 1RM strength in the bench press exercise. I recommend that Coleman et al. alter their statements accordingly.

Counts et al. 2017: https://pubmed.ncbi.nlm.nih.gov/28543604/

*Lines 51-52: I recommend Coleman et al. alter their statements to read somethings along the lines of “To our knowledge, …

*Lines 52-54: As mentioned above, I recommend that Coleman et al. use one term (i.e., “deload” or “detrain”) consistently throughout the manuscript to make it an easier read. If the authors choose to keep the statements about fatigue accumulating, then I emphasize that they provide sufficient clarification, with appropriate references, as to what exactly is accumulating. The statement about re-sensitizing muscle to anabolic stimuli is fine, but I recommend Coleman et al. at least acknowledge (i.e., limitations section) that they have included no such measurements of anabolic signalling.

Discussion:

*Lines 323-324: I would argue that Ogasawara et al. 2011 and 2013 actually examined the effects of deloading on muscular adaptations prior to Coleman et al.’s study. It might also be argued that Vann et al. did too, albeit investigators did not include a subsequent period of training. I recommend Coleman et al. alter their statement to be more accurate.

*Lines 324-327: I recommend that Coleman et al. alter their statements as they did not find minimal evidence, but rather no evidence to suggest that a 1-week deload had any influence of hypertrophy, endurance, or countermovement jump performance. Coleman et al. also did not include any measurements of anabolic signalling, which therefore limits the ability to tell whether there even was a “re-sensitization,” yet alone a beneficial effect of re-sensitization. It can indeed be stated that there appeared to be no beneficial effects of including a 1-week deload, but I do not believe that a discussion on re-sensitization can be had/is appropriate without including such measurements.

*Lines 327-328: I recommend Coleman et al. clarify what “modest benefit” means, and further alter their statements to appropriate reflect their data. Coleman et al. state, “Conversely, the traditional training group experienced modest benefits in measures of both isometric and dynamic strength.”, which could be suggestive that the detraining group did not increase strength. In actuality, however, the deload and traditional training groups both demonstrated increases in 1RM and isometric strength, albeit with evidence (i.e., a small effect size) that the change in isometric (but not 1RM) strength favored the traditional training group.

*Lines 332-333: Using percentage change values for muscle thickness arguably carries a limitation in the sense that the percentages may not scale accurately towards/at the extreme ends of each range (i.e., for measurements of smaller versus larger muscles; please see to the Atkinson and Batterman 2012 article for an in depth explanation). I recommend Coleman et al. avoid discussing the percentage, or magnitude of change in muscle size in the absence of a time matched, non-exercise control, and simply state that they documented similar pre- to post-changes in muscle thickness across all measurement sites.

Atkinson and Batterman 2012: https://pubmed.ncbi.nlm.nih.gov/22760546/

*Lines 338-344: I have a hard time with the notion that the Ogasawara et al. 2011 and 2013 studies are not ecologically valid. They just were set up to answer a different question than Coleman et al.’s study. I recommend Coleman et al. alter their statements accordingly.

*Lines 345-347: I recommend Coleman et al. acknowledge that they included no objective measures of fatigue or anabolic signalling. Because the notion that fatigue can dissipate suggests that something is accumulating, I recommend Coleman et al. also clarify what exactly would accumulate or alter/remove such statements.

*Lines 347-348: Coleman et al. can also reference the work of Hortobágyi et al. 1993, wherein similar feelings were reported (i.e., lazy, sluggish) by participants who ceased training for 14 days (please see to the Hortobágyi et al. manuscript, specifically their discussion).

*Lines 348-354: I generally understand the belief that employed a reduced training volume deload rather than complete cessation of training may help avoid feelings of lethargy. The Vann et al. 2021 study (referenced in Coleman et al.’s introduction) did not examine feelings of lethargy, but found no evidence that a 1 week period of active recovery (i.e., reduction in training volume) resulted in any differences in muscle soreness compared to passive recovery (i.e., complete cessation from training). I recommend Coleman et al. consider the results of Vann et al. prior to suggesting that a reduction in training volume could have resulted in different outcomes and then acknowledge that their statements are purely speculative. I am not certain the current data provided by Coleman et al. give any indication to suggest that using reduced training volume rather than complete cessation of training can elicit superior hypertrophy. In addition, I again recommend Coleman et al. clarify what they mean by the dissipation of fatigue and add that it is speculative to be making these statements (i.e., because no objective markers of fatigue were measured).

*Lines 356-362: I recommend Coleman et al. adjust their statements to align accordingly with their data (i.e., as mentioned previously).

*Lines 362-365: The lack of effect that deloads had on strength adaptations in previous literature (i.e., Ogasawara et al. 2011 and 2013) might be due to the investigators measuring 1RM strength every 3 weeks (i.e., repeated exposure to the 1RM resulted in a “training stimulus”). I recommend Coleman et al. read a paper by Spitz et al. 2020, and/or Morton et al. 2016 (both support the statement about repeated exposure to 1RM possibly being a training stimulus). I recommend Coleman et al. highlight the fact that previous work retested strength very frequently, and that their current design did not, which might explain these discrepancies.

Spitz et al. 2020: https://pubmed.ncbi.nlm.nih.gov/33017302/
Morton et al. 2016: https://pubmed.ncbi.nlm.nih.gov/27174923/

*Lines 369-370: I recommend that for clarity, Coleman et al. provide an example of a resistance training protocols that would be consistent with that of strength athletes. It could be as simple as “i.e., higher % 1RM training.”

*Lines 370-373: I am sure negative effects of deloading on strength were actually observed in the current study. Indeed, the deload training group also experienced increases in 1RM and isometric strength. I recommend Coleman et al. alter their statements accordingly. I also am not certain (or possibly am just confused) as to why there are so many conflicting statements about what is “done in the field” and Coleman et al.’s current design. In the hypertrophy section of the manuscript, Coleman et al. state their design more closely resembles what is done in the field. Please keep things consistent for an easier read, or if I am misinterpreting, I apologize but could use clarification.

*Line 375: I recommend Coleman et al. adjust their statements to accurately reflect their data.

*Lines 376-380: Without a time matched, non-exercise control group, I do not believe it is appropriate to discuss a transfer of strength. I recommend Coleman et al. read a paper by Spitz et al. 2023, wherein the authors quantified (via meta-analysis) the increase in strength for a non-specific strength task. The general conclusion from Spitz et al. is that strength gain can occur in both specific and non-specific strength tests, but the magnitude of change in non-specific strength is seemingly quite small and difficult to capture in a single study (i.e., 326 participants per group would be required for a lower bound estimate of 0.22).

Spitz et al. 2023: https://pubmed.ncbi.nlm.nih.gov/36396899/
*Line 382: I recommended Coleman et al. remove the work “Local” from “Local Strength-Endurance” for consistency and clarity purposes. Or add it into the methods… just keep things consistent.

*Line 383: There does not appear to be any evidence in favor of the traditional group for muscular endurance adaptations. If anything, the mean difference effect size favors the deload training group, but the 95% credible interval crosses zero.

*Line 386: Research on the effects of detraining on local muscular endurance is not as limited as Coleman et al. have made it out to be. It was studied as early as 1970 by Sysler and Stull. I recommended Coleman et al. alter their statements to highlight that other works of literature on the topic of detraining and muscular endurance (e.g., Sysler and Stull, Coratella et al. 2016) have generally not included subsequent cycles of training as did Coleman et al.’s design.

Sysler and Stull 1970: https://pubmed.ncbi.nlm.nih.gov/5266481/
Coratella et al. 2016: https://pubmed.ncbi.nlm.nih.gov/27801598/

*Line 389-390: Although the Haff 2015 textbook is often cited similar to how Coleman et al. have done, I am not certain whether capillarization, mitochondrial activity, et cetera actually play a role in local muscular endurance adaptations, nor am I certain that change in them over a 1-week time period would do anything to influence muscular endurance adaptations. Coleman et al. did not include any of the aforementioned measurements, and previous literature (e.g., Sysler and Stull 1970) has shown that including a 1-week period of inactivity did not result in any negative effects on muscular endurance. Nor did the deload result in negative effects in Coleman et al.’s study. I recommend Coleman et al. alter their statement accordingly.

*Lines 391-392: Increases in 1RM strength indeed play a role in absolute muscular endurance adaptations, but Coleman et al. did not include 1RM strength testing on the leg extension, or a citation for their statement. I recommend Coleman et al. read a paper by Chatlaong et al. 2022 and include it as a citation. Coleman et al. can acknowledge that 1RM was not tested and speculate that due to the high load training protocol employed (i.e., 8-12RM), 1RM strength likely would have increased and resulted in individuals lifting a lower percentage of 1RM post-intervention, which in turn provides a reasonable hypothesis to explainedthe increased local muscular endurance observed for the traditional and deload training groups. Coleman et al. can also consider reporting the change in leg extension total training volume from the initial to final training visit and speculate that an increase in total training volume may reflect an increased work capacity and explain their local muscular endurance adaptations, but I tend to believe most data support that the increase in 1RM strength plays a larger role in absolute muscular endurance adaptations.

Chatlaong et al. 2022: https://scholar.google.com/scholar?cluster=6018601029821481831&hl=en&as_sdt=0,25

*Lines 392-394: I recommend Coleman et al. alter their statements to appropriately reflect their data, as there was no negative effect of a period of detraining on local muscular endurance adaptations. Indeed, the detraining and traditional training groups demonstrated similar increases (i.e., from pre- to post-intervention) in the number to repetitions completed to failure.

*Lines 396-400: I recommend Coleman et al. read a paper by Hackett et al. 2022, wherein the authors examined the effects of total repetitions per set on absolute and relative muscular endurance adaptations. The general conclusion from Hackett et al. is that higher repetitions are only superior to lower repetitions for stimulating local muscular adaptations, but only when investigators employed a relative muscular endurance test. Because Coleman et al. used an absolute muscular endurance test, I am not certain the relevance of their current statements.

Hackett et al. 2022: https://www.sciencedirect.com/science/article/abs/pii/S0765159722000405

*Lines 402-409: I recommend Coleman et al. include more than one citation when discussing the body of literature on muscular power and detraining, and further highlight that the Hortobágyi et al. 1993 reference was a cross-sectional study that did not include a subsequent cycle of training. I am not certain whether it is very relevant to their current design. I also recommend if Coleman et al. are going to discuss adaptations to the stretch shortening cycle, they provide a citation that supports their statement, which suggests that performing resistance training explosively would extend to greater performance on a countermovement jump.

*Lines 402-409: Apologies if I am wrong, but it is my understanding the secondary analyses were not done on the change from mid- to post-intervention. If I am correct, I recommend Coleman et al. please alter their statements accordingly.

*Lines 419: To my knowledge, this is the first time Coleman et al. state that they tried to promote functional overreaching. I would recommend Coleman et al. define what this means, and highlight that previous literature (i.e., Ratamess et al. 2003, which was cited in Coleman et al.’s introduction) also seemingly failed to induce functional overreaching. I think it is quite likely that “overtraining” through resistance exercise alone may be very difficult to induce in humans (i.e., please see to the previously mentioned paper from Kataoka et al. 2021). Indeed, in a study by Corréa et al. 2021, participants performed 144 weekly sets of resistance exercise (across all muscle groups i.e., ~16 per muscle group) to failure and still demonstrated increases in 1RM strength (suggestive that they were not “overtrained”).

*Lines 420-423: Is there any knowledge of whether the protocol employed in the current study was higher volume than what was previously done by participants? In other words, I ask if Coleman et al. have any indication that the individuals’ previous training volumes were lower than what was performed in their current study (my guess is they likely were). It may have some utility in trying to “overtrain” participants. I also recommend Coleman et al. just state they are certain participants performed the lower body, and avoid stating 90 weekly sets for all muscle groups as they cannot be sure that individuals performed the upper body training sessions.


*Lines 424-428: I am not certain what these statements add to the manuscript. Just because participants believed they did not need a break and planned to return to their normal training routine does not inherently mean they did. Nor does it mean that they would continue training in a manner similar to how they had in the current study, which I believe would be more relevant.

*Lines 429-432: I recommend Coleman et al. alter their statements to appropriately reflect their data (i.e., the only difference appeared to be for isometric strength). I also recommend Coleman et al. highlight that there were largely no negative effects of using a deload (i.e., all outcomes increased from pre- to post-intervention). Based on Coleman et al’s data, the use of a 1-week deload after 4-weeks of training can be used if an individual is wishes to, or conversely if the individual does not wish to, he/she seemingly will not experience any negative effects of doing so (other than a less robust increase in isometric strength).

*Lines 432-434: I recommend Coleman et al. also add that it is unknown if deloads would be important with higher training frequency (i.e., targeting each muscle group 6 days per week).

*Lines 437-440: I do not actually believe this is a limitation, nor do I under why exactly can these results cannot be extrapolated elsewhere. These statements more or less seem like fillers.

*Lines 440-443: Because both training groups exercised on the smith squat, I am not sure this is a limitation per se. It could be argued that traditional training group was exposed to the smith squat in two more sessions compared to the detraining group, but obviously this was by design. Please add the word “smith” if statements are to be kept in (I recommend just removing as I do not believe it to be a limitation):

*Lines 443-452: I generally agree here… because individuals performed 10 sets/muscle group I am not sure any subtle differences between participants training to and close failure would have any measurable effect on muscle growth outcomes. Even if it did, groups were randomized and therefore it would seem unlikely that individuals who “ended their sets prior to reaching momentary muscular failure” were disproportionally allocated into one training group versus the other.

*Lines 455-458: I recommend Coleman et al. remove these statements, as it was the purpose of the study (i.e., not a limitation).

*Lines 458-459: I recommend Coleman et al. just discuss their lower body protocol. As mentioned by Coleman et al. in a few lines above 458-459, they cannot be certain how much of the upper body protocol was performed. In addition, I ask what does the word relatively refers to (i.e., other literature, previous volume, et cetera)?

*Line 461: I recommend Coleman et al. remove “sometimes performed by physique athletes and other gym enthusiasts.” They do not seem relevant, and I am not certain if anyone is performing such volumes of exercise to or near failure.

Experimental design

***PLEASE SEE TO WORD DOCUMENT FOR ALL OF MY COMMENTS***

Materials and Methods:

*Lines 82-83: Again, I recommend that Coleman et al. use one term (i.e., “deload” or “detrain”) consistently throughout the manuscript. The current statement presumably could add confusion to the manuscript: “experimental group that detrained (i.e., no RT) for 1 week at the midpoint of a 9-week RT program (DELOAD: n = 25) or a traditional training group that performed the same…”

*Lines 99-101: I recommend Coleman et al. clarify if a metronome was used to standardize the repetition cadence or exactly what “as monitored by the research staff” means.

*Lines 128-130: I recommend Coleman et al. provide a schematic illustration of the experiment in a figure, or clarify which specific measurements were taken on each testing session. e.g., does “separate testing sessions” mean separate for muscle thickness and strength, or just separate from training visits? When were 10RMs completed? What order were measurements taken in (i.e., power, then isometric, then 1RM is how the current manuscript suggests)? et cetera.

*Lines 135-138: Were participants told to refrain for all testing procedures or just anthropometrics. I recommend Coleman et al. clarify, especially with respect to the time of day at which strength testing was conducted, and whether caffeine intake was monitored.

*Lines 145-148: I recommend Coleman et al. clarify the positioning (i.e., supine, standing) of participants for muscle thickness measurements and whether it remained constant in pre- to post-testing.

*Lines 148-151: I recommend Coleman et al. clarify if the same technician who took muscle thickness measurements obtained muscle thickness dimensions, and whether such was done after all data collection was completed (i.e., were pre- and post-muscle thickness dimensions measured obtained after all data collection was terminated, or were pre-muscle thickness measurements obtained at pre-testing). I understand the technician was blinded to all assessments.

*Lines 164-165: Are the ICC and CV values for muscle thickness over a 9-week time frame. If not, I recommend Coleman et al. remove such statements, as it would inform little with respect to the current study design. If they were, then I recommend Coleman et al. please clarify if the ICC and CV values presented are for the same technician who took measurements in their current study, and which specific muscles and muscle sites the ICC and CV values represent (i.e., because the more proximal sites can be more difficult to measure).

*Lines 176-177: Were participants given feedback on their jump height?

*Lines 189-190: Why were 30 seconds of rest allotted between each attempt, and why were 4 trials used? I don’t have any inherent issues with the procedures, assuming it was kept constant between pre- and post-testing. I recommend Coleman et al. note that their lab has previously used these procedures (albeit in the calf muscles) and cite Van Every et al. 2022. I also recommend Coleman et al. clarify if participants were given feedback during isometric strength assessments.

Van Every et al. 2022: https://pubmed.ncbi.nlm.nih.gov/36048793/

*Line 204:. I recommend Coleman et al. clarify when 3 versus 5 minutes of rest was taken, and whether this was consistent between pre- and post-testing. I also recommend Coleman et al. clarify how precise they could be for 1RM assessments (i.e., the smallest load increment for 1RM strength testing).

*Lines 207-208: Similar comments as above (i.e., for muscle thickness ICC and CV values) for the ICC and CV values for the Smith machine back squat. I recommend Coleman et al. clarify if the ICC and CV values are for the Smith machine back squat 1RM, and if they over a 9-week time frame. Because strength assessment has a skill specify component, I am not certain what ICC and CV values inform (i.e., I do not believe they should be used in place of a time matched, non-exercise control group).

*Lines 211: I recommend Coleman et al. provide clarity that they used an absolute muscular endurance assessment (i.e., did not update the load to reflect the individuals current body mass at post-testing). I also recommend Coleman et al. clarify how precise they could be for strength-endurance assessments (i.e., the smallest load increment for strength-endurance testing). Out of curiosity, why was initial body mass used for the assessment? I understand that previous work from your lab (i.e., Plotkin et al. 2022), as well as other literature (i.e., Rhea et al. 2003). Lastly, I recommend Coleman et al. consider altering “strength-endurance” to “local muscular endurance” to better fit with the resistance training literature.

Rhea et al. 2003: https://pubmed.ncbi.nlm.nih.gov/12580661/

Statistical Analyses:

I am completely fine, and strongly agree with Coleman et al.’s statements from Lines 240-243. I recommend, however, that Coleman et al. calculate effect sizes by dividing group differences by the change score variability rather than baseline variability for reasons discussed in a paper by Dankel et al. 2021. I recommend Coleman et al. alter all Results, Figures, Tables, et cetera accordingly.

Dankel et al. 2021: https://pubmed.ncbi.nlm.nih.gov/30358698/

*Lines 254-265: Why were analyses on the readiness to train data performed on mid- and post-intervention assessments rather than the change from mid- to post-intervention? Or am I misinterpreting?

Validity of the findings

***PLEASE SEE TO WORD DOCUMENT FOR ALL OF MY COMMENTS***

Results:

*Lines 267-271: I do not believe the “s” is not needed after “cm” or “kg”.

*Lines 285: Table 1 does not show the posterior probabilities for group differences.

*Lines 286-288: There was no evidence for a difference between groups at any muscle sites. I am not certain what this statement adds to the manuscript.

*Lines 299-311: I recommend Coleman et al. please refrain from referring to the traditional training group (i.e., TRAD) as “control”. Please alter accordingly throughout the manuscript.

*Lines 301-303: I recommend Coleman et al. please clarify exactly how they have reached the conclusion that there was evidence for greater strength adaptations for traditional training group compared to the detraining group, and which strength assessment they are referring to (i.e., 1RM or isometric). The 95% credible intervals for univariate group differences all cross through zero.

*Lines 303-304: Where do these p values come from? They are not the exact values found in Table 3.

*Lines 304-305: Where are the multivariate analyses are displayed in Table 3.

*Lines 305-308: There was no evidence for a difference between groups in muscular endurance adaptations or counter movement jump height. I am not certain what this statement adds to the manuscript. In addition, I strongly recommend Coleman et al. alter their statement (i.e., “standardized mean difference effect sizes showed that if group differences did exist … they may be small to large in favor of control for 1RM and isometric strength.” to appropriately reflect their data. Figure 3 shows that the standardized mean difference effect size for 1RM strength is small, with 95% credible intervals (please clarify that data are indeed surrounded by 95% credible intervals) that range from a small effect to no effect (i.e., it crosses zero) in favor of the traditional training group. The standardized mean difference effect size for isometric strength is also small, but contains a 95% credible interval that ranges from a small effect to mediation effect in favor of the traditional training group. In previous work from Coleman et al.’s lab (i.e., Van Every et al. 2022), the authors state “Rather than interpreting effects from a single test, or set of tests, the results were interpreted on a continuum using all statistical outcomes, in combination with theory and practical considerations.” I recommend that Coleman et al. similarly avoid interpreting and drawing conclusions on the effects of deloading on strength adaptations, as a whole, from a small effect size in favor of the traditional training group in isometric strength (especially in the absence of a time-matched, non-exercise control group which would have accounted for measurement error).

*Lines 317-320: I recommend Coleman et al. analyze the change in readiness to train data from mid- to post-intervention. I tend to think a difference in changes in sleep quality and/or muscle soreness from mid-intervention (i.e., before the deload) to post-intervention between groups would inform much more than a single snapchat at mid- and post-intervention.

I recommend Coleman et al. include the training logs (i.e., load and repetitions lifted) over the first four weeks and then last four weeks. I do not think statistical analyses need to be performed on the data, but I do believe including such would provide a more objective means of assessing whether there appeared to be any indication that individuals should employ a deload. I acknowledge that by design individuals were randomly allocated to the deload group.

Figures, Figure Captions, Tables:

I recommend Coleman et al. avoid using the word “control,” and change it to something along the lines of traditional training group, TRAD, et cetera. throughout.

All figure captions, and data need to be altered once the change in variability is used to calculate effect sizes. In addition, I recommend Coleman et al. highlight that the data points are surrounded by 95% credible intervals in the figures (provided they are… it is difficult to tell exactly what the reflect, apologies).

Table 1:
- Please remove the “s” in “kgs”, “cms”.
- Please refer to jump as counter movement jump for consistency.
- Please refer to muscular endurance as “strength-endurance” for consistency.

Table 2:
- Please note these are for the changes pre- to post-intervention in ___.
- Please put in the title that Table 2 also included muscle growth outcomes (or morphological outcomes i.e., however it is termed throughout the manuscript).
- Please remove the “s” in “mms”.

Table 3:
- Please note these are for the changes pre- to post-intervention in ____.
- There are no multivariate analyses included in Table 3, or it is mislabeled. Please fix.
- Please refer to endurance as “strength-endurance” for consistency. Please also just use (repetitions as the unit).

Conclusion:

I strongly recommend Coleman et al. alter their statements to appropriately reflect their data (i.e., isometric strength was the lone outcome to show a difference). I also recommend Coleman et al. to add a statement about what their data mean for researchers (i.e., should a 1-week deload be an important methodology consideration in resistance training studies?).

Additional comments

***PLEASE SEE TO WORD DOCUMENT FOR ALL OF MY COMMENTS***

Thank you for the opportunity to review Coleman et al.’s manuscript. The authors sought to investigate the effects of a brief deload on lower body muscular adaptations in resistance trained individuals. They compared changes in lower body muscle size and strength, as well as local muscular endurance and muscle power adaptations between two training programs that differed namely with respect to the inclusion of a 1 week deload (at the midpoint of the 9-week period of training). I read the pre-print a couple weeks ago and have made my way through the submitted manuscript a few times now. I would like to applaud the conciseness of the authors’ writing style and congratulate the researchers on their efforts to carry out such an experiment. I do not believe the current version of the manuscript is suitable for publication and suggest major revisions. If clarifications and adjustments are to be made, then I believe the manuscript will be great and important addition to the resistance training literature. I have provided detailed comments below. I acknowledge my comments are lengthy, and in some cases a bit nit-picky. My hope is that the authors critically think on them, and provide clarifications/adjustments where they see fit.

Annotated reviews are not available for download in order to protect the identity of reviewers who chose to remain anonymous.

Reviewer 2 ·

Basic reporting

no comment

Experimental design

no comment

Validity of the findings

no comment

Additional comments

no comment

Annotated reviews are not available for download in order to protect the identity of reviewers who chose to remain anonymous.

---

## Round 0.2 · Major Revisions

While you have addressed most of the comments from the first reviewer, the second reviewer feels you have not taken their constructive criticisms on board sufficiently and is now recommending Reject. I therefore encourage you to respond to the final request from the first reviewer but even more so to look back at the initial comments from the second reviewer and see how you can better address their concerns.

Reviewer 1 ·

Basic reporting

Please see to PDF document.

Experimental design

Please see to PDF document.

Validity of the findings

Please see to PDF document.

Additional comments

Please see to PDF document.

Annotated reviews are not available for download in order to protect the identity of reviewers who chose to remain anonymous.

Reviewer 2 ·

Basic reporting

no comment

Experimental design

no comment

Validity of the findings

no comment

Additional comments

no comment

---

## Round 0.3 · Minor Revisions

Thanks for attending to most of the comments from the reviewers. Some small ways in which the manuscript needs to be revised include the following three small points. Note: all line numbers refer to the track changes word document.

Line 30: while you added in Kataoka as suggested, you still appear to have Vieira in the reference. Or is this a different paper from Vieira?

Line 527: the first reviewer requested a definition of potentiating effect here but even though you replied you have clarified with a definition I could not see one. Please update this request.

Reviewer 2 mentioned the inconsistencies in the use of the term deload and detraining. While I see many more examples of deload in the manuscript, there are still many examples of detraining as well. Please check to see if this can be further addressed.

Reviewer 1 ·

Basic reporting

Adaptations is spelled incorrectly in the abstract.

Experimental design

N/a.

Validity of the findings

N/a.

Additional comments

I would like to thank Coleman et al. for addressing the majority of my comments and providing rebuttals where appropriate. I would also like to commend the authors and researchers for the ability to conduct such an experiment. I believe the manuscript has been much improved since the original pre-print, and is now suited to be a great addition to the literature. Congratulations.

---

## Round 0.4 · accepted · Accept

Thanks for attending to the comments of the reviewers and I.